# Study on Scheme of Outbound Railway Container Organization in Rail-Water Intermodal Transportation

**Jiahao Zhao**, **Xiaoning Zhu \*** and **Li Wang**

School of Traffic and Transportation, Beijing Jiaotong University, Beijing 100044, China;
zhaojiahao@bjtu.edu.cn (J.Z.); liwang@bjtu.edu.cn (L.W.)

\* Correspondence: xnzhu@bjtu.edu.cn

**Abstract:** It has been proven that exploring how to achieve an efficient transportation system is a crucial component of every sustainable transportation study. Rail-water intermodal transportation is recognized as one of the future transportation methods for being efficient, economical and environmentally friendly. To improve the efficiency, reduce transportation costs and maximize the resource utilization of outbound intermodal container transportation, based on the relationship between the container central station and the port station in the actual problems, the organization of railway container transportation was studied. A multi-objective optimization model was established in order to minimize the total cost in the process of transportation, which means maximizing the resource utilization and ensuring it is environmentally friendly. Additionally, an improved genetic algorithm (GA) was developed to solve the model. The calculation results of the model are obtained by the simulation calculation. The comparison with the conventional fixed axis transportation organization method proves that the model and algorithm can reduce costs by up to 24.57%. The result also shows that the container transport organization should be tried to satisfy the direct loading and discharging condition of "train-ship," meanwhile reducing the storage time at the high toll central station. In conclusion, the model and algorithm are feasible and effective. Due to the universality of the model, it can be easily used and generalized in or out of China.

**Keywords:** rail-water intermodal transportation; outbound railway container; integrated transportation system; railway operation and management; genetic algorithm

## 1. Introduction

Container transportation refers to the mode of transportation using containers as carriers. The intermodal transport of goods with containers as basic transport units has become the development trend of international trade transportation. Rail-water intermodal transportation is an important part of it, which is a seamless transportation mode for rail and water transportation. It has gradually attracted everyone's attention under the trend of economic globalization and sustainability. Compared with traditional transportation, rail-water intermodal transportation with the characteristics of safety and environmental protection, low transportation costs, and large-scale special trains have great significance for the development of container transportation.

Take Europe as an example, because of the technical and economic advantages of railways in terms of transportation capacity, transportation efficiency, emission levels, and ease of congestion, intermodal transport is considered a major contender for road freight in mid- to long-distance corridors [1]. Europe has gradually established a relatively sustainable and collaborative intermodal transport service network with containers as the main cargo-carrying unit and railways as the core. At the same time, the main ports in Europe have been built the container railway station. Railways play an important role in container collection and distribution in the ports.

Europe advocates that railways should take more transportation volumes, which requires an increase in the service level of the multimodal transport chain and increases the market share of railway transportation in terms of medium- and long-distance freight. Specifically, before the end of 2020, they aim to establish a framework system for the management and payment information system of the EU multimodal transport; before 2030, they will require 30% of road freights to be transferred to rails and waterways, and will reach 50% by 2050 [2]. In 2017, the proportions of rail-sea intermodal transportation were 36% in Duis, 39% in Hamburg and 47% in Bremen.

However, the development of multimodal transport is different in China. Although it has been more than ten years of promoting the mode of rail-water transportation in China, due to insufficient transportation capacity of the railway system and other reasons, the implementation of rail-water transportation has been slow.

With the continuous advancement of the "Thirteenth Five-Year Plan" and the "Belt and Road" strategy, China has established the notion that lucid waters and lush mountains are invaluable assets and acted with resolve and intensity as never before to strengthen environmental protection. At the same time, an aim of energy consumption per unit of GDP (Gross National Product) falling more than 20% has been set. As the pillar industry of the country, the transportation industry will play a vital role in the process.

That's why China is entering a golden period of development of transportation infrastructure, improvement of service level, and transformation and development. It has been two years since the first China-Europe rail-water intermodal train was officially launched from Chongqing on 28 December 2017. As the volume of cargo increases, the gap compared with developed countries and regions is gradually emerging, especially in terms of rail-water transportation. In China's main coastal ports, railways account for less than 5% of the port collection and transportation volume, while in the world's typical iron-water combined transport ports, such as Hamburg, Bremen, and Long Beach ports, the proportion exceeds 35% [3].

Compared to "rail-road" and "water-road" intermodal transportation, the organization of "rail-water" intermodal transportation involves more efficient coordination of the operating organization between the completely independent railway and waterway systems. The coordination degree of the "rail-water" intermodal transportation is largely restricted by the railway's collection and distribution organization. Therefore, this is an urgent problem to be studied.

In this paper, an outbound railway container organization problem in rail-water intermodal transportation is studied. Figure 1 is a typical layout of China's rail-water combined port [4]. There are two types when the location of the railway central station is considered. In one type, for example, Ningbo Port, the railway central station is located next to the dock front and part of the port yard at the terminal is used as a railway yard. The other, for example, Dalian Port, the railway central station is located outside of the port gate. For outbound railway containers, the difference is the distance from the train to the ship. In this question, since the specific operation time and process are not involved, the distance will be uniformly expressed in terms of transportation costs.

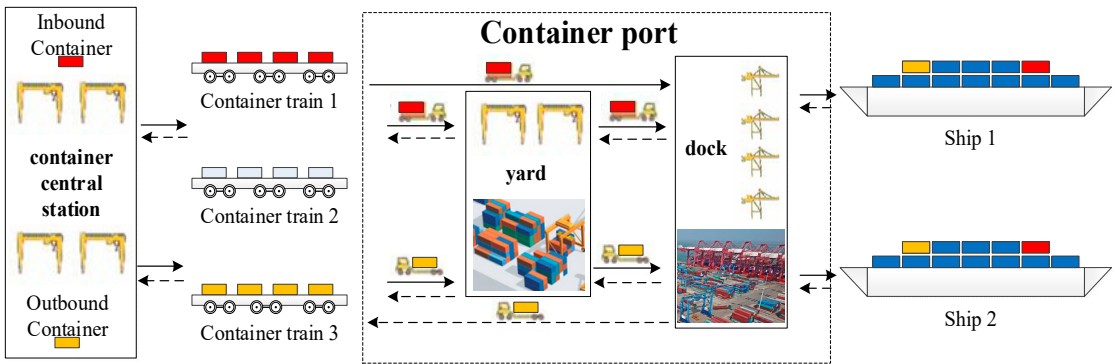

**Figure 1.** Sketch map of the container transportation of rail-water intermodal transportation.

In China, according to the port operation manual, the railway container transportation process between the container central station and the port station is mainly divided into three parts:

(1) Cargos are concentrated at the central station. When technical conditions, such as the full axis, meet the eligibility requirements, the technical loading and unloading operations are carried out. The train departs to the port station.

(2) After arriving at the port, the "train-ship" direct loading and unloading conditions will be considered by the arrival time and container volume. Then different operations will be carried out on different containers based on the results.

(3) The cargo loading operations will be executed at the dock of the port. After that, the ship will depart the port according to the schedule, as shown in Figure 1. The red boxes represent the containers transported by container trains to the container port. Then they are transported in the port by container trucks. Finally, the whole process ends with the loading operation at the front of the dock.

It can be seen from Figure 2 that when targeting the minimum transportation cost, the container central station operation, railway transportation operation and container port operation need to be fully considered. Therefore, a railway container transportation organization optimization model based on rail-water transportation was established to determine the train operation plan for the container central station. The main contributions of this problem were:

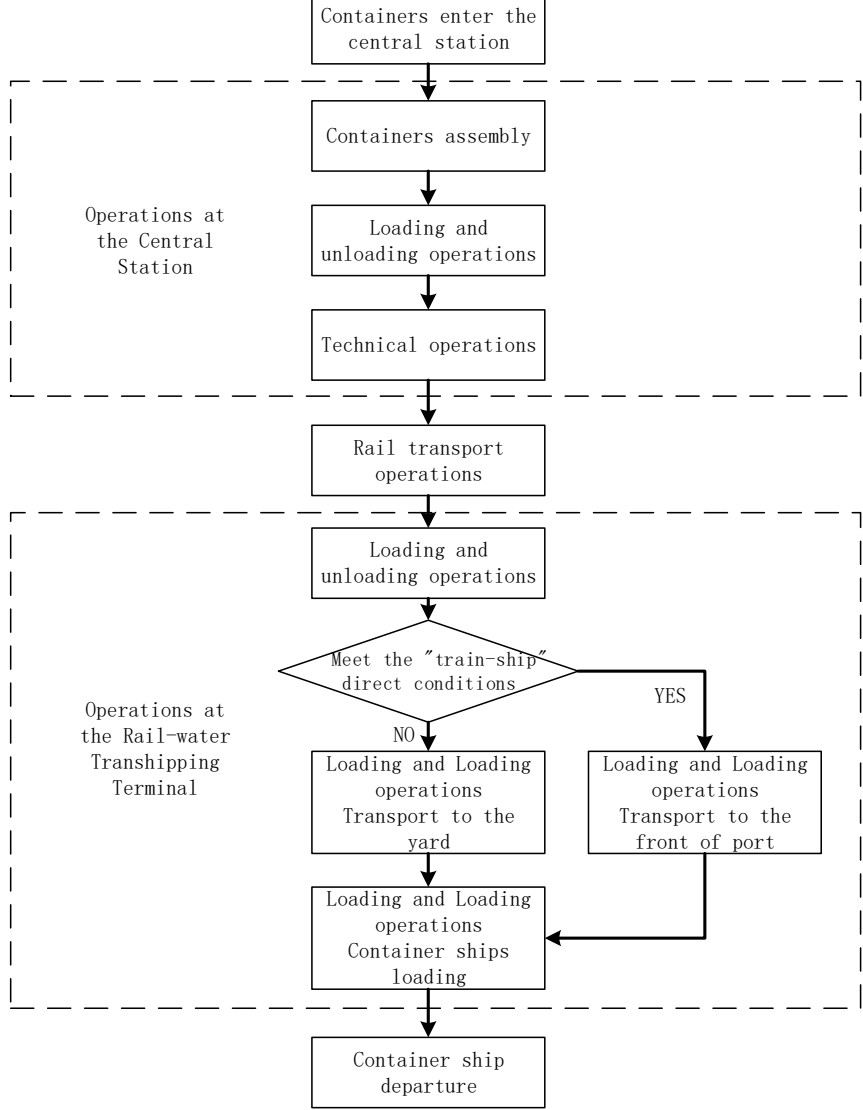

**Figure 2.** Operation flow chart of container transportation of rail-water intermodal transportation.

(1) On the basis of summarizing the domestic and international research on container hot metal transport according to the development status and needs of China's hot metal rail transport, we considered the constraints of train and ship intermodal transport, railway transportation time limit, port or station operating capacity and other constraints.

With the goal of minimizing total transport costs which means maximizing the resource utilization and ensuring it is environmentally friendly, an organization model for the consolidation of rail-water transport containers under the constraints of time and space was established.

(2) The model comprehensively considered some practical problems, such as different numbers of attracting containers for different containers, and different costs for different operations after railway containers arrive at the port, making the model closer to the actual application.

(3) Design the corresponding algorithm to solve the model, verify the model and algorithm based on the relevant data, and finally prove that the model algorithm is effective.

In this paper, literature on intermodal transport, transportation planning optimization problems and coordination between railway station and port are discussed in Section 2. In Section 3, the model formulations, basic assumptions and objective function are proposed. Then in Section 4, an improved genetic algorithm (GA) was developed to solve the model. In Section 5, a simulation experiments on the model and algorithm was carried out to test the feasibility of the algorithm. The comparison and analysis of related results are shown in the Section 6. At last, conclusions are summarized in Section 7.

## 2. Literature Review

The study of intermodal transportation has been discussed for years. In 2004, Macharis and Bontekoning believed that multimodal transport would become an emerging field of transportation research and summarized the operations research models and modeling issues used in this field [4]. As one of the most recent review papers on multimodal transportation planning problems, Crainic, Teodor Gabriel, and Kap Hwan Kim discussed the definition, transported by a sequence of at least two transportation modes, and the importance of intermodal transportation in international commerce [5]. Then, the research in the area of multimodal transport planning accelerated. SteadieSeifi and Maryam gave a structural overview of the multimodal transport literature since 2005 [6]. Intermodal freight transport has been discussed for decades as an alternative to unimodal road transport. However, it still does not represent a significant portion of the total freight market. In 2016, Behdani, Fan, Wiegmans, and Zuidwijk gave a detailed description of this integrated view for synchromodal freight transport, which is based on the design and operation of intermodal transport. A mathematical model for designing service schedules for synchromodal freight transport systems was also presented [7]. Zhang and Pela developed a model that captures relevant dynamics in freight transport demand and supply, flexible multimodal routing with transfers and transshipments and enables comparative analysis of intermodal and synchromodal operations [8]. In 2019, Ambra, Caris, and Macharis introduced a computational movements model to assess intermodal and synchromodal resilience [9]. Giusti, Riccardo, et al. surveyed the existing scientific literature concerning synchromodal logistics and provided the concept of a new coordination platform [10].

Back to the outbound railway container organization in rail-water intermodal transportation, many published papers related to intermodal transportation are based on determining the mode choice [11,12] and or hub location problem [13–16]. Considering the intermodal transportation organization problem, a literature review can be divided into three groups:

### 2.1. Intermodal Transportation Network Channel and Path Problems

There is quite a lot of research on multimodal transport networks and routes. A mathematical model was developed for the design of a multimodal transport radial network with multiple stakeholders and multiple types of containers by Qiang, and Wang [17]. Beuthe used a certain area as an example to transport 10 different types of goods in a multimodal transport network and developed a traffic demand allocation model that comprehensively considered each feasible route and the transportation

mode. The combination of direct and cross demand elasticity coefficients is presented for the three modes of transportation: railway, highway, and water transportation [18]. Gelareh, Nickel, and Pisinger proposed a mixed integer programming formulation for hub-and-spoke network design in a competitive environment [19].

Bock and Stefan proposed a new real-time-oriented control approach in order to expand load assembly, reduce empty vehicle trips, and handle dynamic disturbances [20]. Chang established a multi-objective, multi-commodity flow problem with time windows and concave costs according to the characteristics of the international multimodal transportation route selection problem and used relaxation and decomposition techniques to decompose the original problem before proceeding to the optimization study [21]. Asvin proposed a transportation model that combines transportation and routing and shows how visibility in transit can be used to adjust transportation plans relative to known states of the transportation system [22]. Athanasios et al. proposed a path optimization algorithm for multimodal transportation networks and verified the effectiveness and complexity of the algorithm in response to the delays in multimodal transportation processes and cross operations [23]. Baykasoğlu, Adil, and Kemal Subulan presented a mixed-integer mathematical programming model for a multi-objective, multi-mode and multi-period sustainable load planning problem [24].

In the context of the "Belt and Road" initiative, Sun and Jiasen designed an effective two-way auction mechanism for the procurement of intermodal transportation services [25]. Vasco analyzed the advantages and disadvantages of the combination of railway and highway, waterway, aviation and other modes of transportation, proposed a transfer technology when the mode of transportation was changed for efficient multimodal transportation services, and carried out energy consumption of railway transportation [26]. Fan and Jin proposed a performance-driven multi-algorithm selection strategy for energy consumption optimization of sea-rail intermodal transportation [27]. Hilde Heggen provided decision support for routing customer orders throughout the intermodal network with the aim of minimizing total transport costs and maximizing capacity utilization [28].

*2.2. Transportation Planning Optimization Problems*

Next, the research on the planning optimization problem is also quite rich. White et al. used transportation planning theory to carry out detailed optimization allocation research on railway container transportation [29]. Newman studied the rail transport links of multimodal transport containers, that is, the daily and weekly plans of direct and non-direct trains and how the containers were placed on the multimodal rail transport trains [30]. Cao studied the optimization of railway container transport plans under multimodal transport, analyzed the characteristics of multi-cycle plans, and established a large-scale plan with the goal of maximizing the total profit generated by all registered cargo transportation in multi-plan cycles [31]. She also proposed a random integer programming model that considers the demand and supply of railway container transportation under random demand and aims to maximize the expected profit under uncertain demand. The model was solved and the effectiveness of the algorithm was verified by numerical experiments [32]. Danloup, Allaoui, Gonzalez-Feliu, and Goncalves proposed a new model for the pickup and delivery problems with transshipment and time windows. A transportation network with two transportation modes, road and sea, was considered to study the role of transshipment. They applied the model to a three agri-food companies case study which showed that the total cost and the total amount of $CO_2$ emissions were reduced by using collaboration and transshipment [33].

The railway transportation organization has been made to optimize the train scheme [34–38]. Yang clarified that the research on transportation organization between railway container yards plays an important role in guiding the design, construction and operation of railway container yards [34]. By analyzing and referring to existing cargo flow path optimization models, a multi-objective 0–1 planning railway container path optimization model was developed. Miller, aiming at the reliability of the railway container transportation network, established a two-level stochastic programming model of cost and service level, and solved it by taking the American railway container multimodal

transportation network as an example [35]. April established a train selection model based on multi-commodity network traffic with the goal of minimizing operating costs to determine train loading and unloading plans [37]. Liu and Deng proposed a multimodal transport network that considered freight consolidation through the cooperation of freight forwarders. In the actual case on the "China Railway Express," five different transportation demand scenarios were tested [39]. Corry and Kozan proposed optimization models for foreign railway container vehicle loading and transportation organizations, respectively, to optimize the flow path, marshalling plan, and operation chart of railway container trains [40].

### 2.3. Coordination Optimization between Ships and Trains

The last section is research on how to optimize rail-water coordination better. Zhao briefly introduced the coordination area and related concepts. Then an inbound container distribution organization model was established considering many factors in order to minimize the total container-hours in the coordination area [41]. Wu studied the optimization of yard operations and developed a linear mixed-integer programming model for scheduling different types of equipment and planning storage strategies in an integrated manner [42]. Hartmann, Sönke discussed the scheduling of reefer mechanics at container terminals. Reefer mechanics plug and unplug reefer containers such that due times are met [43]. Li, Na, et al. addressed the land-side disruption where truck arrivals deviate from their schedule and proposed a response mechanism that maintains the high resilience ability [44]. Ku and Arthanari proposed a stochastic dynamic programming model to calculate the minimum expected number of shuffles for a pile of containers with a departure time window [45].

The above research is only for the research of railway container intermodal transportation. Although Zhang [46] et al. have carried out some research on the organization of iron and steel intermodal transportation and container trains of iron and steel intermodal transportation, they have paid more attention to the study of the theory of railway transportation organization instead of comprehensive consideration of the impact of water transport.

## 3. Model Formulation

### 3.1. Basic Assumptions

The rail-water terminal transportation process has many influencing factors, including a wide range of time and space complexity. After analyzing the problem, in order to better describe the problem, several assumptions are made and explained in this section before the mathematical model is established.

(1) The containers involved in the problem are all international TEU (Twenty-feet Equivalent Unit); all trains operate in the form of direct container trains which are the highest transportation efficiency;

(2) For customers, the loading and unloading costs of each container are implemented in accordance with relevant regulations, which have nothing to do with other factors;

(3) In China, all outbound containers departure stations are container central stations. They are located at the center of the railway network where there are enough empty vehicle resources. Therefore, the container specific vehicle bottom and empty vehicle constraints are not considered;

(4) The cargo assembly process of the cargo at the container center station meets a uniform distribution, and the assembly time of each container is $t_{ass}$.

### 3.2. Symbol Notations

Some symbolic notations used in inbound container distribution organization model are defined as follows. The following are the necessary parameters:

- $n$: the serial number of containers in chronological order at each container central station
- $T$: the set of unit time interval during the planning cycle

- $I$: the set of container liner ships in a planning cycle
- $i$: the serial number of the container liner ship, $i \in I$
- $J$: the set of container trains in a planning cycle
- $j$: the serial number of the container trains, $j \in J$
- $K$: the set of container central stations in a planning cycle
- $k$: the serial number of the central stations, $k \in K$
- $NS_j$: the total number of containers on container liner ship $i$
- $NT_j^k$: the total number of containers on container train $j$ at container central station $k$
- $NTS_j^k$: the total number of direct loading and discharging containers on container train $j$ at container central station $k$
- $NTY_j^k$: the total number of containers transported to yard on container train $j$ at container central station $k$
- $NT_{\max}$: the maximum number of containers on a container train
- $NT_{\min}$: the minimum number of containers on a container train
- $NY_{\max}^{cenk}$: the maximum number of containers allowed in the operation zone of container central station $k$
- $NY_{\max}^{port}$: the maximum number of containers allowed in the yard of port
- $N_0^{cenk}$: the total number of remaining containers at container central station $k$ during the previous planning cycle
- $N_0^{port}$: the total number of remaining containers at the port during the previous planning cycle
- $M$: a large integer value
- $L^k$: the distance between container station $k$ and the port
- $V^k$: the average travel speed of the trains departed from container central station
- $T_n$: the moment when the container $n$ arrives at the central station
- $T_{kj}^{lea}$: the departure moment of container train $j$ from central station $k$
- $t_n$: the residence time of container $n$ at the central station, $t_n = T_n - T_{kj}^{lea}$
- $T_{kj}^{arr}$: the moment when the container train $j$ arrives at the port
- $T_i^{lea}$: the moment when the container liner ship $i$ departs from the port
- $T_{\min}$: the minimum arrival time interval of container trains
- $t_{ass}$: the accumulation time for a unit container
- $cost_{sto}^{cenk}$, $cost_{sto}^{port}$: the storage price of containers in container central station $k$ and port storage yard
- $cost_{t\text{-}y}^{cenk}$, $cost_{y\text{-}tr}^{cenk}$: the loading and unload container operation charge of "truck-yard", "yard-train" in container central station $k$ and port storage yard, CNY per TEU
- $cost_{tr\text{-}t}^{port}$, $cost_{t\text{-}sh}^{port}$, $cost_{t\text{-}y}^{port}$, $cost_{y\text{-}t}^{port}$: the loading and unload container operation charge of "train-truck", "truck-ship", "truck-yard", "yard-truck" at the port, CNY per TEU
- $cost_{tra}^{tra}$, $cost_{veh}^{tra}$, $cost_{time}^{tra}$: distance fee of the train travel, CNY per train per kilometer; distance fee of the vehicle travel, CNY per vehicle per kilometer; cost of travel time, CNY per train per hour
- $cost_{tr\text{-}sh}^{port}$, $cost_{tr\text{-}y}^{port}$, $cost_{y\text{-}sh}^{port}$: the container transport charge of "train-ship", "train-yard", "yard-ship" at the port, CNY per train per TEU

The following are the decision variables for the container:

$$Dec_{kij}^n = \begin{cases} 1, & \text{when container } n \text{ is transported from station } k \text{ to port by train } j. \\ 0, & \text{else} \end{cases}$$

$$Dec_i^t = \begin{cases} 1, & \text{when the departure time of ship } i \text{ is } t \\ 0, & \text{else} \end{cases}$$

$$Imp_{kj}^n = \begin{cases} 1, \text{ when container } n \text{ is "train-ship" dircet transported from station } k \text{ to port by train } j. \\ 0, \text{ else} \end{cases}$$

Among those, $Dec_{kij}^n$ is used to connect containers with trains and ships; $Dec_i^t$ aims to connect departure time with ships; $Imp_{kj}^n$ represents the "train-ship" direct transportation situation at the port.

### 3.3. Objective Function

In order to reduce the cost of containers at the railway center station, it is necessary to run container trains as quickly as possible, and the cost of rail transport is inversely proportional to the number of containers loaded by container trains; at the same time, the train schedule must take into account the schedule of the port station. The objective function of the model of the problem can be expressed as:

$$\text{Min}Z_1 = \text{Min } Cost^{cen} \tag{1}$$

$$\text{Min}Z_2 = \text{Min } Cost^{tra} \tag{2}$$

$$\text{Min}Z_3 = \text{Min } Cost^{port} \tag{3}$$

(1) $\text{Min}Z_1 = \text{Min } Cost^{cen}$. This denotes the goal of minimizing the total cost of the container at the container center station, whose cost includes container storage fees, loading and unloading operation fee, etc. Therefore

$$Cost^{cen} = Cost_{sto}^{cen} + Cost_{han}^{cen} = Cost_{sto}^{cen} + (Cost_{unl}^{cen} + Cost_{load}^{cen}) \tag{4}$$

where $Cost_{sto}^{cen}$ based on different arrival times is shown in Figure 3. As the storage cost of the central station is charged according to the specific time of arrival of different containers, the storage cost for different containers is different.

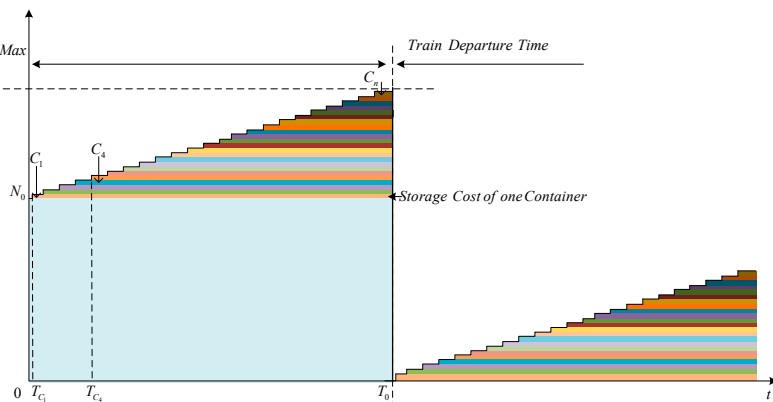

**Figure 3.** Schematic diagram of calculation on total cost in container central station.

As shown in Figure 3, because the cargo assembly process meets a uniform distribution, the storage cost for each container could be formulated as follow.

$$Cost_{C_1\text{-}sto}^{cen} = cost_{sto}^{cen} \cdot t_{C_1} = cost_{sto}^{cen} \cdot (T_0 - T_{C_1})$$

$$Cost_{sto}^{cen} = cost_{sto}^{cen} \cdot \sum_n \sum_{i \in I} \sum_{j \in J} Dec_{ij}^n \cdot t_n = cost_{sto}^{cen} \cdot \int_0^t f(t)dt$$

$$= cost_{sto}^{cen} \cdot \int_{N_0}^{t'} f'(t')d(t') = F'(t')\Big|_{N_0}^{t'} = cost_{sto}^{cen} \cdot [F'(t') - F'(N_0)] \tag{5}$$

When each train leaves the central station, the remaining containers are automatically counted as the initial number of containers for the next train $N_0$. Then

$$
\begin{aligned}
Cost_{sto}^{cen} &= cost_{sto}^{cen}\cdot\left\{ \begin{array}{l} \left[t' + (t' - t_{ass}) + (t' - 2t_{ass}) + \cdots + (t' - \left\lfloor \frac{t'}{t_{ass}} \right\rfloor\cdot t_{ass})\right] \\ -\left[\frac{N_0}{t_{ass}} + (\frac{N_0}{t_{ass}} - t_{ass}) + (\frac{N_0}{t_{ass}} - 2t_{ass}) + \cdots + (\frac{N_0}{t_{ass}} - \left\lfloor \frac{N_0}{t_{ass}{}^2} \right\rfloor\cdot t_{ass})\right] \end{array} \right\} \\
&= cost_{sto}^{cen}\cdot\left\{ N_0\cdot t + \left[t + (t - t_{ass}) + (t - 2t_{ass}) + \cdots + (t - \left\lfloor \frac{t'}{t_{ass}} \right\rfloor\cdot t_{ass})\right]\right\} \\
&= cost_{sto}^{cen}\cdot\left[ N_0\cdot t + (t - \left\lfloor \frac{t'}{t_{ass}} \right\rfloor\cdot\frac{t_{ass}}{2})\cdot(\left\lfloor \frac{t'}{t_{ass}} \right\rfloor + 1)\right]
\end{aligned}
\tag{6}
$$

The loading and unloading operations of the container center station are divided into two parts: loading and unloading of trucks and loading and unloading of trains. Due to the low proportion of direct loading and unloading at the container center station, the operation process of all containers was unified in the question. The loading and unloading operation was divided into two steps. The first step was to unload the goods to the main operation area after the goods arrive. The second step was to load the truck from the yard to the container train.

$$
Cost_{han}^{cen} = Cost_{t\text{-}y}^{cen} + Cost_{y\text{-}tr}^{cen} = (N_0 + n)\cdot(cost_{t\text{-}y}^{cen} + cost_{y\text{-}tr}^{cen})
\tag{7}
$$

(2) $MinZ_2 = Min\ Cost^{tra}$. This represents the goal of minimizing the total cost of all shipping cost during the transportation process. Here the transportation cost includes the technical operation cost and the vehicle use cost.

$$
Cost^{tra} = Cost_{tra}^{tra} + Cost_{veh}^{tra} + Cost_{time}^{tra} = cost_{tra}^{tra}\cdot L + cost_{veh}^{tra}\cdot NT_j\cdot L + cost_{time}^{tra}\cdot\frac{L}{V}
\tag{8}
$$

(3) $MinZ_3 = Min\ Cost^{port}$. This represents the goal of minimizing the total cost of all containers at the port station. Here the cost includes container storage fees, loading and unloading operations fees, etc., then

$$
Cost^{port} = Cost_{sto}^{port} + Cost_{han}^{port} + Cost_{tra}^{port} = Cost_{sto}^{port} + (Cost_{unl}^{port} + Cost_{load}^{port}) + Cost_{tra}^{port}
\tag{9}
$$

At the same time, considering different operations of containers under different conditions of the station, the operation of the port station was divided into two types.

(1) The arrival time of the train meets the "train-ship" direct loading and unloading needs. The container arriving at the port will be directly transported by the truck to the front of the terminal for loading.

The operation of the port station is specifically: train-truck loading and unloading operation, truck-traveling operation, truck-truck loading and unloading operation. As it is a direct loading and unloading operation, $Cost_{sto}^{port} = 0$, then

$$
Cost^{port} = \sum_{j}\sum_{n} Imp_j^n\cdot(cost_{tr\text{-}t}^{port} + cost_{t\text{-}sh}^{port} + cost_{tr\text{-}sh}^{port})
\tag{10}
$$

(2) The arrival time of the train doesn't meet the "train-ship" direct loading and unloading needs. Arrival containers will be transported by container trucks to the terminal yard and will be transferred to the front of the terminal for loading at the beginning of the loading operation.

The operations at the port station include train-truck loading and unloading operations, truck-truck moving operations, truck-truck loading and unloading operations, truck-truck moving operations, truck-truck loading and unloading operations.

$$
\begin{aligned}
Cost^{port} &= \sum_{j}\left(NT_j - \sum_{n} Imp_j^n\right)\cdot cost_{sto}^{port} + \sum_{j}\left(NT_j - \sum_{n} Imp_j^n\right)\cdot(cost_{tr\text{-}t}^{port} \\
&+ cost_{t\text{-}sh}^{port} + cost_{t\text{-}y}^{port} + cost_{y\text{-}t}^{port}) + \sum_{j}\left(NT_j - \sum_{n} Imp_j^n\right)\cdot(cost_{tr\text{-}y}^{port} + cost_{y\text{-}sh}^{port})
\end{aligned}
\tag{11}
$$

In summary, in order to solve this problem, the calculating procedure of weight was simplified. The problem is formulated as

$$MinZ = Min(\alpha Cost^{cen} + \beta Cost^{tra} + \chi Cost^{port}) \tag{12}$$

*s.t.* (subject to):

$$N_0^{port} + \sum_0^t \sum_{j\in J} NT_j \cdot Dec_i^t \geq \sum_0^t \sum_{i\in I} NS_i \cdot Dec_i^t, \forall t \in T \tag{13}$$

$$N_0^{port} + \sum_j NT_j \geq \sum_i NS_i \tag{14}$$

$$NT_{min} \leq NT_j \leq NT_{max} \tag{15}$$

$$(T_{j+1}^{lea} - T_j^{lea}) \cdot t_{ass} \geq N_0^{cen} + NT_{min} \tag{16}$$

$$0 \leq N_0^{cen} + \left\lceil \frac{t}{t_{ass}} \right\rceil - \sum_0^t \sum_{j\in J} NT_j \cdot Dec_i^t \leq NY_{max}^{cen} \tag{17}$$

$$0 \leq N_0^{port} + \sum_0^t \sum_{j\in J} NT_j \cdot Dec_i^t - \sum_0^t \sum_{i\in I} NS_i \cdot Dec_i^t \leq NY_{max}^{port} \tag{18}$$

$$(T_j^{lea\text{-}k} + \frac{L^k}{V^k}) - (T_j^{lea\text{-}k} + \frac{L^k}{V^k}) \geq T_{min}, \forall t \in T, \forall j \in J \tag{19}$$

$$T_i^{lea} \geq \left[ T_j^{lea} + \frac{L}{V} + \sum_n Imp_j^n \cdot (t_{tr\text{-}t}^{port} + t_{t\text{-}sh}^{port}) + (NS_i - \sum_n Imp_j^n) \cdot \binom{port}{tr\text{-}t} + t_{t\text{-}sh}^{port}) \right] \cdot \sum_n Dec_{ij}^n \tag{20}$$

$$\sum_{i\in I} \sum_{j\in J} Dec_{ij}^n = 1, \forall j \in J, \forall i \in I \tag{21}$$

$$\sum_{i\in I} Dec_i^t \leq 1, \forall t \in T, \forall i \in I \tag{22}$$

Equation (12) is the objective function of the entire model, representing the least cost of the entire process; Equations (13) and (14) represent container flow constraints. The sum of all containers arriving at the port and the initial container at the port must not be less than the demand of the ship; Equations (15) and (16) represent the operating conditions of container trains. The trains must operate at full axles and must not exceed the maximum number of containers allowed; Equations (17) and (18) mean that the container center station operation area and the container port yard should not exceed the maximum allowable stock at any time; Equation (19) means that the capacity of the train bound for the container port station is limited by the line capacity, so the time interval between the receiving and departing trains at the container port station cannot be less than $T_{min}$; Equation (20) says that the sailing time of the ship should be sufficient to complete the train-ship loading and unloading and handling operations; Equations (21) and (22) represent the unique constraint of containers, that is, each container can only be transported by one train and one ship, and trains and ships have only one departure time.

## 4. Solution Algorithm

The determination model of the container iron-water combined transport scheme is a multi-objective optimization model. Due to the conflicting objectives in the problem, the complexity of the problem greatly increased. For such models, the commonly used solving algorithms include heuristics such as a genetic algorithm and simulated annealing algorithm [47–49].

In order to solve the whole model based on the characteristics of the above model, this paper draws on the fixed cost transportation problem based on the genetic algorithm, further explores the coding method and the construction method of the selection, crossover and mutation operator to solve

the model. The specific steps of the algorithm are as follows. The flow chart is shown in Figure 4. The detailed algorithmic steps were described as follows:

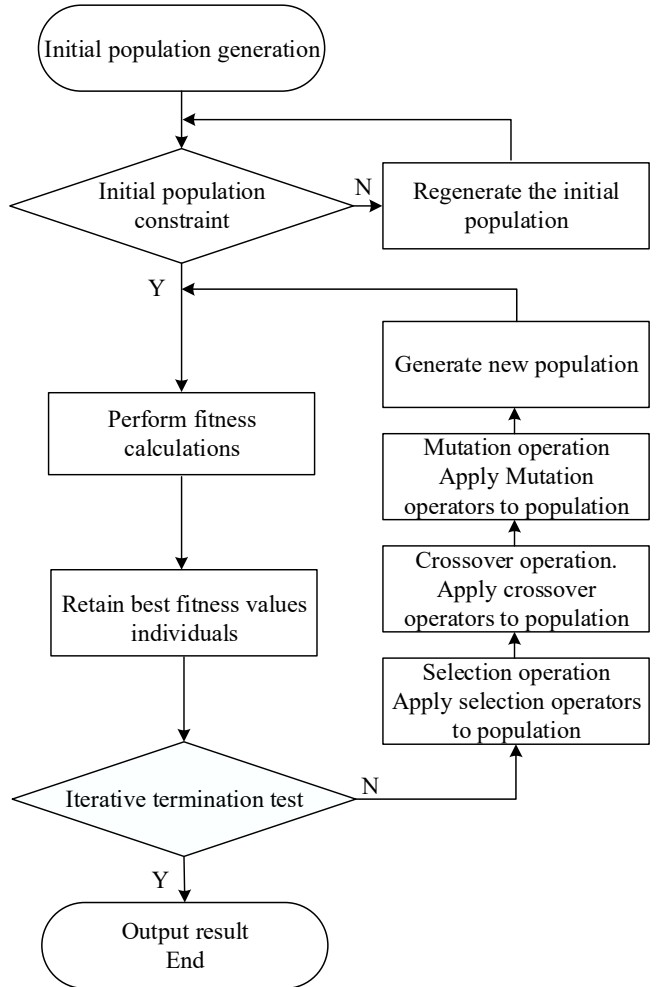

**Figure 4.** Improved genetic algorithm procedure.

Step 1: Design of Chromosome Code

The container train transport plans are encoded in the algorithm. Each transport plan corresponds to a chromosome which contains the train operating time, the number of trains and the container ship information corresponding to the container on the train.

Firstly, a chromosome was created as shown in Figure 5. The length of the chromosome represents the calculation time of the model, where each gene represents a unit of time. Next, the train's source central stations are randomly generated and saved in the corresponding time unit gene based on the train constraints and number of container trains in the planning period; at the same time, 0 is filled in the moment when the train has not arrived. So far, we have obtained an initial transport scheme for the container train.

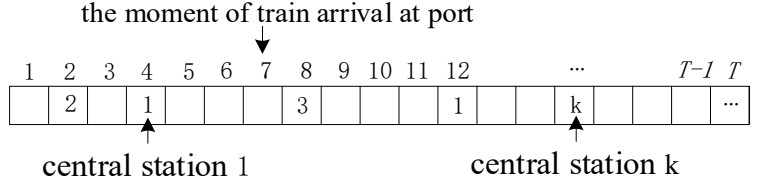

**Figure 5.** Schematic diagram of chromosome.

In order to meet specific algorithm requirements, the [m, n] matrix is used here to represent the result.

At the same time, a new variable $Dec_{kj}^t$ is added to connect train departure time with the container trains.

$$Dec_{kj}^t = \begin{cases} 1, & \text{when the departure time of train } j \text{ from station } k \text{ is } t \\ 0, & \text{else} \end{cases}$$

Step 2: Algorithm Initialization

All relevant data needs to be entered. Besides the population size *Size*, the termination number of iterations *Ite*, crossover probability $p_{cro}$ and mutation probability $p_{mut}$ are given.

Initial transport plans are generated according to the above chromosome coding method. If it is found that every chromosome meets all conditions after the constraint tests, the chromosome is regarded as a primary chromosome. If not, the initial transportation scheme is repeatedly generated until a feasible primary chromosome is obtained. After getting the primary chromosomes, the initial population is obtained: *Gen* = 1.

Step 3: Fitness calculation

The fitness of each independent individual is calculated in the population; the chromosomes that have been generated are decoded.

First the departure time of the train at the cargo central station is calculated according to the time of the arriving train; the maximum number of train containers is obtained by operating time constraints. In the meantime, the total cost of each individual is calculated by the timetable of departure information.

In the cost minimization problem studied, the smaller the function value, the higher the fitness. Therefore, the fitness function in this paper is expressed by the inverse of the objective function, specifically:

$$f = \frac{1}{Cost_{all}} = \frac{1}{Cost^{cen} + Cost^{tra} + Cost^{port}} \tag{23}$$

Step 4: Selection

In order to save the good individuals (high fitness) in the population and copy them to the next population, an improved algorithm based on roulette (the higher the fitness, the greater the probability of being selected) was proposed. The size of the cross section in the roulette is proportional to the value of the fitness function of each chromosome. The larger the value, the larger the cross section. Besides, the optimal individual preservation rule is added based on roulette, which is used to select better individuals to accelerate the search speed. At last, according to the corresponding selection probability, the selection operator is executed to generate the next generation population.

Step 5: Crossover

In order to speed up the discovery of better solutions and prevent convergence to local extreme points at the beginning of the iteration, a uniform crossover method is adopted in this algorithm.

A randomly generated genetic mask string of the same length as the individual is used to determine how the offspring individuals obtain the gene. The specific steps are as follows.

A parent *V* is selected from the population and a random number $e \in (0, 1)$ is generated. If $e < p_{cro}$, then we select the chromosome as a parent. Parents in the population are paired randomly, and the two parents who have been paired successfully are crossed as parent 1 and parent 2.

Position-based crossover is used during the crossover operation. Two different gene positions are randomly generated. Offspring 1 inherits the gene fragments between the intersections of parent 2 and the remaining genes inherit the non-replicated genes in parent 1 in order; offspring 2 inherits the gene fragments between the intersections of parent 1 and the remaining genes inherit the non-replicated genes in parent 2 in order, shown in Figure 6.

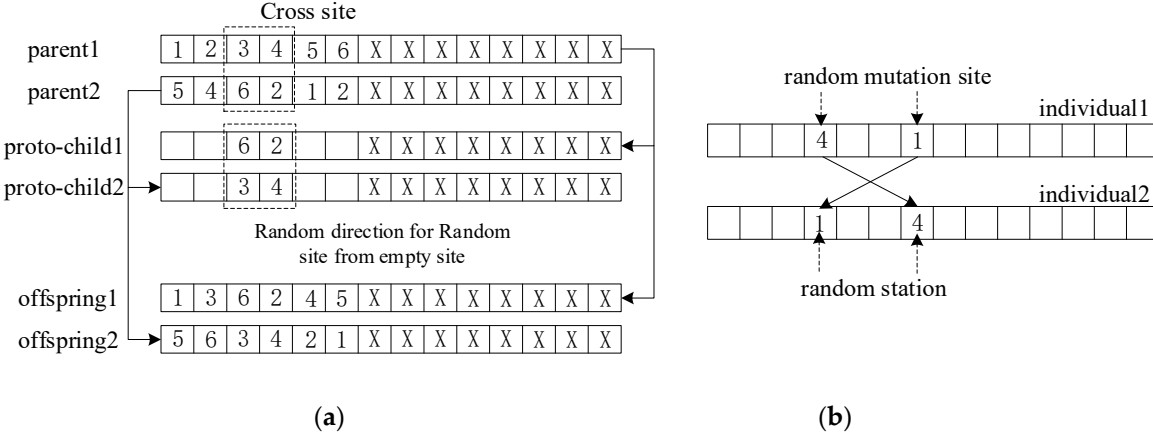

**Figure 6.** (**a**) Schematic diagram of crossover operation; (**b**) Schematic diagram of mutation operation.

Step 6: Mutation

The algorithm will mutate the chromosome according to the mutation probability to prevent immature convergence and accelerate convergence. The specific steps are as follows. First, each individual in the corresponding population generates a random number, and if the individual corresponds to that, a mutation operation is performed on the individual. In this paper, the basic mutation method is used for mutation operation. Two gene positions are randomly determined, and genes at the two positions are exchanged, shown as Figure 6. The generated offspring chromosomes are checked for legality and illegal individuals are legalized. Repeat the above process several times to get a new population.

All heuristic algorithms apply two methods for searching, local search and global search. Local search is to find the optimal solution in the neighborhood. The corresponding ones are selection operator and crossover operator, which try to find the best people in the tribe. If there is only local search, it is easy to fall into the local optimal solution. The result of the algorithm must be to find the global optimal solution. This requires jumping out of the local search. This is the reason why mutations are needed.

Step 7: Termination

If the program meets the set termination conditions, the algorithm is terminated and the optimal result will be output. If not, step 3 will be repeated until the set termination conditions are met.

## 5. Number Experiment

In order to test the feasibility of the algorithm, this paper carried out simulation experiments on the model and algorithm. First, we selected four container central stations and one port station for a one-week study. The relevant data is shown in Table 1.

**Table 1.** Basic data of container central station.

| Name of Station | Distance from the Port (km) | Operation Ability (TEU/Y) | Daily Send Amount (TEU/D) | Goods Attraction Speed (TEU/h) | Railway Expense (CNY/TEU) | Calculated Expense (CNY/TEU) | Running Time (h) |
|---|---|---|---|---|---|---|---|
| ZZ1 | 1223 | 360000 | 187 | 1.5 | 3911 | 2049.7 | 15 |
| WH2 | 983 | 450000 | 234 | 2 | 3428.5 | 1713.7 | 12 |
| XA3 | 1620 | 510000 | 265 | 2 | 4936.2 | 2605.5 | 20 |
| CQ4 | 2128 | 430000 | 224 | 1.5 | 6625.2 | 3316.7 | 26 |

ZZ1: ZHENGZHOU; WH2: WUHAN; XA3: XIAN; CQ4: CHONGQING.

Among them, the daily average number of sent containers were calculated by the actual survey data to be about 0.52‰ of the operating capacity. At the same time, considering the relationship between the specific location of the railway container station and the delivery to the destination port station, the cargo volume of the designated port station was obtained. Here, due to the small amount of cargo, the storage capacity constraints of the station were no longer considered.

Rail freight was obtained from the 12306 website (China Railway Customer Service Center) by entering the OD (Origin-Destination) point; we calculated the freight by calculating the freight rate published on the 12306 website; train running time was calculated according to 80 km/h and rounded up; as the unit of each part of the objective function was the same, the value of the weight coefficient was 1, and the values of other data can be divided into three parts as follows, as described below.

China Railway Customer Service Center is one of the most important windows for railway service customers. It integrates passenger and cargo transportation information across the country and provides passenger and cargo transportation services and public information inquiry services for social and railway customers.

Container railway central station: loading and unloading operations cost 195 CNY/TEU, handling costs 45 CNY/TEU, over-scale and other costs are 30 CNY/TEU, storage costs are 75 CNY/TEU per day. If the arrival of the box is within 24 h, there is no charge; more than 24 h and less than 48 h is recorded as one day. At the same time, due to the certainty of the transportation plan, the last box of the central station loading and unloading can be directly loaded and unloaded, and the operation time is neglected.

Rail transportation: Considering that the exact operating cost of the train cannot be accurately calculated, here it was calculated based on the inquiry price of the 12306 Website, and the increase in the number of containers after the vehicle is full can give a certain discount. Specifically, when the number of vehicle groupings reaches the full axle requirement of 40, a price discount of 0.5‰ is provided for each additional vehicle bottom. That is, when the grouping number reaches a maximum of 50, the freight has a 5 ‰ discount.

Ningbo Port (rail-water transshipping terminal): the train-truck loading and unloading operation costs 195 CNY/TEU, truck-traveling operation costs 49.5 CNY/TEU, truck-truck loading and unloading operation costs 150 CNY/TEU, truck-truck loading and unloading operation costs 490 CNY/TEU. At the port station yard from the fifth day, a storage fee of 10 CNY/TEU per day is charged from that day. It is assumed here that the "train-ship" direct loading and unloading requires a fixed time to complete. When the train arrives 6–12 h earlier than the sailing time of the schedule, the "train-ship" direct loading and unloading can be performed. The information of liner ships in Ningbo Port is shown in Table 2.

**Table 2.** Information of liner ships in Ningbo Port.

| Arrival Moment (h) | 0 | 59 | 92 | 142 | 190 |
|---|---|---|---|---|---|
| Arrival Inbound Containers (TEU) | | 290 | 276 | 380 | 382 |
| Demand for Outbound containers (TEU) | | 250 | 300 | 350 | 400 |

## 6. Results

In this paper, the population size, crossover probability, mutation probability, and evolution algebra of the genetic algorithm were set at 100, 0.8, 0.1, and 300, respectively.

In order to verify the feasibility and efficiency of the model and algorithm, two sets of actual operational data, "80TEU" and "100TEU" of the fixed-axis principle were involved. At present, container transportation of Chinese railways uses the fixed-axis operation principle. In other words, the number of containers on one train is the same; the container train will be launched when the number of assemblies reaches the set full-axis standard. This method has the advantages of being easy to operate and simple, so it is widely used in practice.

"80TEU" and "100TEU" in the vertical axis represent the standard of the fixed-axis principle. 80TEU in the vertical axis means the container train departs when exactly 80 containers have been loaded. Similarly, 100TEU in the vertical axis means the container train departs when exactly 100 containers have been loaded. This paper first calculates the fixed-axis operation principle by computer, and the results are shown in Table 3.

**Table 3.** Results of the model and fixed-axle method.

| | Container Volume (TEU) | Shipping Cost (CNY) | Shipping Cost per Container (CNY/TEU) | "Train-Ship" Container (TEU) | Total Cost (CNY) | Cost per Container (CNY/TEU) |
|---|---|---|---|---|---|---|
| Model | 1567 | 7536206 | 4809.3 | 1497 | 7550171 | 4818.2 |
| 80TEU fixed-axis | 1200 | 6979265 | 5816.1 | 80 | 7202705 | 6002.3 |
| 100TEU fixed-axis | 1200 | 6904804 | 5754.0 | 600 | 7024504 | 5853.8 |

In Table 3, the cost result of 80 TEU and 100 TEU fixed-axis are calculated. We could see that the container volumes are the same at 1200 TEU. Although the shipping cost per container for 80TEU and 100TEU are nearly the same, the cost per container of 80TEU is 2.5% higher, which is 148.5 CNY/TEU. The major differences between them are the amount of "Train-ship" containers. It shows that significant cost reductions can be achieved by increasing the "train-ship" rate. Besides, the "train-ship" could save a handle progress from the yard to the ship, which is maximizing the resource utilization and is environmentally friendly.

In order to verify the feasibility and efficiency of the model and algorithm, the results will be compared with the 80TEU and 100TEU fixed-axis operation model. It can be seen that the average container cost optimized by the model and algorithm saves 1184.0, 1035.5 CNY/TEU, and optimizes 24.57% and 21.49%. It shows that the model can effectively improve the efficiency of railway transportation and save transportation costs.

This further illustrates that the restrictions on operations, sailing, and ship schedules will affect the transportation plan, and proves the feasibility and effectiveness of the model and algorithm. The specific results are shown in Tables 4 and 5, and the corresponding central stations and the change in container volume at the port station are shown in Figure 7.

**Table 4.** The result of best chromosome in genetic algorithm.

| From | 2 | 1 | 3 | 4 | 2 | 3 | 1 | 4 | 2 | 3 | 1 | 4 | 2 | 3 |
|---|---|---|---|---|---|---|---|---|---|---|---|---|---|---|
| Central station | WH | ZZ | XA | CQ | WH | XA | ZZ | CQ | WH | XA | ZZ | CQ | WH | X |
| Arrival moment (h) | 32 | 47 | 50 | 66 | 80 | 85 | 111 | 130 | 130 | 135 | 159 | 178 | 180 | 184 |
| Arrival amount (TEU) | 100 | 100 | 90 | 100 | 96 | 80 | 80 | 100 | 100 | 100 | 88 | 96 | 100 | 98 |

**Table 5.** Optimal scheme of container intermodal transportation.

| Central Station | ZHENGZHOU | | | | WUHAN | | | XIAN | | | | CHONGQING | | |
|---|---|---|---|---|---|---|---|---|---|---|---|---|---|---|
| Departure moment (h) | 32 | 96 | 144 | 20 | 68 | 118 | 168 | 30 | 65 | 115 | 164 | 40 | 104 | 152 |
| Arrival moment (h) | 47 | 111 | 159 | 32 | 80 | 130 | 180 | 50 | 85 | 135 | 184 | 66 | 130 | 178 |
| Arrival amount(TEU) | 100 | 80 | 88 | 100 | 96 | 100 | 100 | 90 | 80 | 100 | 98 | 100 | 100 | 96 |

From Tables 3 and 4, the number of containers on most trains is greater than 90TEU. It can be seen that for Z1 ($Cost^{cen}$) and Z2 ($Cost^{tra}$) in the multi-objective function, Z2 accounts for most of the total cost and has a large impact on the results.

At the same time, we can see that the higher the freight rate of the central station, the larger the number of vehicle formations, but there are also some vehicle formations that have just hit the full axle. This is due to the relatively fixed transportation cost; whether it is possible to do direct loading and unloading also has a great impact on the cost of the entire transportation process.

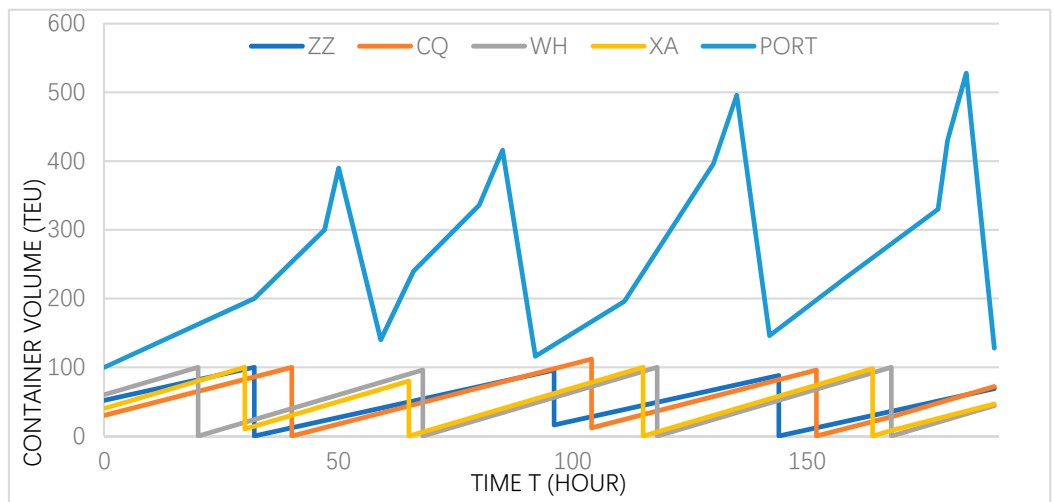

**Figure 7.** Volume change of containers in central stations and port.

## 7. Conclusions

This article studies the organization problem of railway container assembly based on intermodal rail-water transportation. From maximizing the resource utilization and environmentally friendly perspective with the object of minimizing the cost in the assembly process, a railway container transport organization model is established, considering the location of different central stations and attracting goods. Specific practical situations such as ability and then the improved the genetic algorithm solved the model. Finally, through simulation calculation, the model and algorithm were proved to be feasible and effective. According to relevant results, the requirements for improving resource utilization and environmental protection can be effectively met, which will play a positive role in China's sustainable development. Although this article is based on China's national conditions, the principle of universality has been maintained in the process of model establishment. Therefore, for the popularization and application of the model, only the relevant data needs to be replaced and modified. However, as this study is currently considering one-way transport, the vehicle resources at the port were not considered. Future research could focus on two-way transport where more thorough research on empty vehicles is needed. Future work could also include practical application of large-scale central stations and ports.

**Author Contributions:** The authors confirm contribution to the paper as follows: Conceptualization, X.Z. and L.W.; data curation, J.Z., X.Z. and L.W.; formal analysis, X.Z.; funding acquisition, X.Z.; methodology, J.Z. and L.W.; project administration, X.Z. and L.W.; resources, X.Z.; validation, J.Z.; writing—original draft, J.Z.; writing—review & editing, J.Z., X.Z. and L.W. All authors have read and agreed to the published version of the manuscript.

**Funding:** This research received no external funding.

**Acknowledgments:** This work was supported by the Funds of Beijing Jiaotong University (Grant no. 2018YJS073), the National Key R&D Program of China (grant number 2018YFB1201403).

**Conflicts of Interest:** The authors declare no conflict of interest.

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
