# Peer review of "Study on Scheme of Outbound Railway Container Organization in Rail-Water Intermodal Transportation"

_sustainability, doi:10.3390/su12041519_

Round 1

Reviewer 1 Report

Title:

Study on Scheme of Outbound Railway Container 3 Organization in rail-water intermodal transportation

The authors of the article describe rail-water intermodal transportation that is recognized as one of the future transportation methods for its efficient, economical and environmentally friendly character. On the basis of the relationship between the container, central station and the port station, the organization of railway container transportation was studied. A multi-objective optimization model was established from the perspective of the customer in order to minimize the total cost in the process of transportation.

I appreciate the coverage and potential contribution of the paper, but I find a significant need for some correction of the study. Before any future submission for publication, however, it would be good for the authors to consider the following comments.

Comments:

The title clearly reflects the content of the paper.

Abstract

The abstract is well written. I appreciate the coverage and potential contribution of the paper but I find a significant need for clarifying the purpose of the paper. It should reflect what the research purposes were.

Keywords – properly chosen.

The introduction doesn’t clarify the article’s interior.

So in my opinion -  the introduction section should be adjusted to the requirements – maybe the plan of the article should be put in the scheme or something.

Methodology

Methodology part is missed at all. I am sure it should be added

The literature review is in problem description I guess – it should be adjusted to the requirements including more literature sources.

Results

Results and conclusion sections are valuable and show rather properly the articles assumption as well as contribution.

My general remark is related to the formal part of the article – some bullet without content, variable spaces, etc.

Author Response

Reply to Review Reports

Dear Reviewer,

Thank you for your letter and for the reviewers’ comments concerning our manuscript. Those comments are all valuable and very helpful for revising and improving our paper, as well as the important guiding significance to our researches. We have studied comments carefully and have made corrections which we hope meet with approval. The formatting of the references is revised.  

Please see the attachments for the replies to the comments.

Reviewer 2 Report

See enclosed document.

Author Response

(The authors gave the same response as above.)

Reviewer 3 Report

The paper is a classical OR paper, with a contribution difficult to assess in terms of relevance. Moreover, it is difficult to me to state on the suitability of the paper since sustainability issues are not really presented. However, I think that if authors make important changes, it can be suitable for publication. Therefore, I recommend revisions.

Here are my comments:

p. 1, lines 39-42, please add a reference to the statement. In the last statistics I read (but they can be old), Germany's intermodal transport was about 20%, but France and Italy had lower rates. Then, precise if it is rail-road transport, sea-road transport, river-road transport, air-road transport or all those types (or if there are systems with more than two modes included in the statistics).

p. 3., lines 104-121, please consider adding Danloup et al. (2020), since their multimodal planning method considering environmental criteria seems novel to me.

The end of the literature sectionneeds a better motivation of the research. Indeed, the fact water is not considered in one previous work is not an enough reason to motivate the paper: why water is important as mode? What adds to rail or road that makes it competitive and then justifies considering such systems? I think that if the paper allows generalizing on the suitability of intermodal systems (not only rail-road) this is a good motivation, but authors need to justify and state it clearly.

In problem description, how Figures 1 and 2 are defined? Are they extensions/adaptations of existing work or are they authors' elaborations? In the first case, sources need to be cited, in the second, the observation/data collection protocols that allowed defining in a rigourous ways those figures need to be explained (5-10 lines).

The optimization model is a time minimization one, and not a cost (in terms of monetary issues) one. Please, change Cost by Time to make a more coherent model. Moreover, a model is presente but neither discussed nor used in the optimization process. This is a classical issue in problem-solving, algorithmic-based OR research but I remain convinced that the interest of a model is to use and/or discuss it. Please, either discuss the model in terms of solving issues (is it an NP-hard? how can it be solved? Is it close to other problems in litterature? how this type of problems are solved in litterature?) and motivating the use of heuristics or proposing a set of instances and solve them with a commercial solver, then compare those results to the heuristics' ones (same instances) before presenting the real case results. Anyway, if a model is proposed, it is important to provide an order of magnitude of instances that can be solved with this model (either because there are statements of this type in literature after other authors having solved similar models or from test instances proposed and solved by the authors of this paper with a commercial solver).

Personnally, I think a genetic algorithm is not the most fast and robust algorithm to solve this type of problems, since it is well-known that genetic algorithms are good to widely explore solution spaces but are slower to converge than greedy-based, Tabu search or ALNS ones. Why this type of algorithm is chosen? I think the paper should improve by comparing a genetic algorithm with a GRASP or other basic metaheursitic (like simulated annealing, for example).

Results finally are of pseudo-applicative nature, but the algorithms suitability is not addressed in the paper (see my previous comment). Moreover, it is difficult to me to see how data is constructed. Where comes the database from? Is demand a real one or an estimated one? Are stations real? Please, provide a detailed description of how demand and network characteristics are obtainted, and who (in practice) validated them.

Last (but most important), sustainability is not addressed in the paper. I agree that intermodality (if using electical engines, which is not the case in all contexts, for example, many Italian companies use, in some parts of the network diesel locomotives) can lead to a decrease of environmental nuisances, but this is not enough. As said in Danloup et al. (2020) there are ways of estimating environmental (and in a less wide scale, social) impacts from travelled distances and times. Please, provide in discusion an environmental assessment/discussion (or both an environmental and social discussion) to address sustainability issues. Moreover, the case is a Chinese one, very specific (only one case for all China), so how the model/method/results can be generalized, to China, to Asia and worldwide?

I think the paper can be improved so I encoueage authors to do so.

Suggested references:

Danloup, N., Allaoui, H., Gonzalez-Feliu, J., & Goncalves, G. (2020). Assessing the Environmental Impacts of Green Collaboration in Land-Sea Freight Transport. In Handbook of Research on Interdisciplinary Approaches to Decision Making for Sustainable Supply Chains (pp. 113-139). IGI Global.

Author Response

(The authors gave the same response as above.)

Round 2

Reviewer 1 Report

Authors took all remarks under consideration and corrected the paper that is more valuable now. I accept the paper in present form.

Author Response

Dear Editor and Reviewer, Thank you for your letter and for the reviewers’ comments concerning our manuscript.

Reviewer 2 Report

See enclosed document (round 2).

Author Response

Thank you for your letter and for the reviewers’ comments concerning our manuscript. Please see the attachment for the response.

Reviewer 3 Report

The paper is a reviewed version of an already submitted paper. Authors tried to make efforts to improve it, and although epistemologically I do not agree with all elements, I think hte paper has been improved and therefore can be considered for publication.

Author Response

Thank you for your letter and for the reviewers’ comments concerning our manuscript.